# Reference Values for Isometric, Dynamic, and Asymmetry Leg Extension Strength in Patients with Multiple Sclerosis

**DOI:** 10.3390/ijerph17218083

**Published:** 2020-11-02

**Authors:** Kora Portilla-Cueto, Carlos Medina-Pérez, Ena Monserrat Romero-Pérez, José Aldo Hernández-Murúa, Claudia Eliza Patrocinio de Oliveira, Fernanda de Souza-Teixeira, Jerónimo J González-Bernal, Carolina Vila-Chã, José Antonio de Paz

**Affiliations:** 1Institute of Biomedicine (IBIOMED), University of León, 24071 León, Spain; cora1995port@gmail.com (K.P.-C.); japazf@unileon.es (J.A.d.P.); 2Sciences Health School, University Isabel I, 09003 Burgos, Spain; 3Division of Biological Sciences and Health, University of Sonora, Hermosillo 83000, Mexico; 4Escuela Superior de Educación Física, Universidad Autónoma de Sinaloa, Culiacán 80040, Mexico; aldohdez80@hotmail.com; 5Department of Physical Education, Federal University of Viçosa, 36570-000 Viçosa, Brazil; cpatrocinio@ufv.br; 6Exercise and Neuromuscular System Research Group, Superior School of Physical Education, Federal University of Pelotas, 96010-610 Pelotas, Brazil; fteixeira13@hotmail.com; 7Department of Health Sciences, University of Burgos, 09001 Burgos, Spain; jejavier@ubu.es; 8Polytechnic Institute of Guarda, 6300-559 Guarda, Portugal; cvilacha@ipg.pt; 9Research Center in Sports Sciences, Health and Human Development (CIDESD), 5001-801 Vila Real, Portugal

**Keywords:** strength, resistance training, multiple sclerosis, asymmetry, 1RM, maximal voluntary isometric contraction, reference values

## Abstract

Having recognized the value of resistance training in patients with multiple sclerosis (PwMS), there are a lack of lower limb normative reference values for one repetition maximum (1RM) and maximal voluntary isometric contraction (MVIC) in this population. Hence, the purposes of this study were to provide reference values for 1RM and MVIC of knee extensors in PwMS across the disability spectrum and to examine knee extension strength asymmetry. Three hundred and ninety PwMS participated in the study, performing MVIC and 1RM tests of bilateral (both legs together at once) and unilateral (each leg singly) knee extensors. There was no difference in 1RM according to the disease course of MS, but there was according to the degree of neurological disability, being more preserved in those with a lower degree of disability. MVIC tends to be higher in patients with relapsing–remitting MS respect those with progressive MS, and in patients with lower levels of neurological disability. Asymmetry above the values considered normal in 1RM was present in 20–60% of patients and 56–79% in the MVIC test, depending on the type of MS and tended to be lower in those with less disability. Reference values are given by quartiles for 1RM, MVIC, and asymmetry.

## 1. Introduction

Multiple sclerosis (MS) is a demyelinating disease of the central nervous system that most often debuts in young adults [1]. The clinical evolution is usually accompanied by neurological deficits with cognitive, sensory, and physical manifestations, such as muscle weakness [2]. Impairment of mechanical muscle function can have important consequences in persons with multiple sclerosis (PwMS), with more functional impact in the lower extremities [1,3]. Moreover, balance and gait may be affected [4].

In the last two decades, physical exercise has generated a lot of interest as a coadjuvant therapy to improve or maintain functional capacity, cognitive functions, and social activities and, hence, the quality of life [5].

The World Health Organization published in 2010 the “*Global recommendations on physical activity for health*” address for general and healthy populations [6], which is similar to the recommendations for PwMS: do at least 150 min per week of moderate-intensity aerobic exercise or 75 min of vigorous exercise, and perform at least two weekly sessions of musculoskeletal strengthening exercise that involves resistance training [6,7].

In recent years, many studies have included resistance training in their physical exercise programs for PwMS [4,8,9,10], and have shown promising results for improving quality of life and functional capacity after 8 to 12 weeks of training [11]. To increase efficiency as well as to increase exercise safety, training loads should be individualized to each person [12].

There are different types of muscular actions: concentric (muscle length is shortened), eccentric (muscle length is extended), isometric (muscle length is fixed), isotonic (muscle tension is fixed), isokinetic (muscle shortening or lengthening velocity is fixed), and isoinertial (muscle contraction encounters a fixed external resistance), and in all of them muscular tension is generated, which we call force [13]. In daily activities, the most common type of muscle action used is isoinertial, and there are different ways of showing isoinertial muscle strength (e.g., maximum force, muscle power, and force resistance) [13]. Traditionally, resistance training is targeted to improve maximum dynamic strength, and the main consensus established on strength training for health, recommends programming the training load based on each person’s one repetition maximum (1RM) [4,14]. 

The evaluation of muscle strength assessment in addition to being useful to individually prescribe training loads, as well as to categorize individual scores based on the reference values of the population segment to which that person belongs (age, sex, MS type, degree of affectation). However, health and exercise professionals find difficulties in his work for the lack reference data on muscle strength for PwMS with different evolutionary types of the disease or for different degrees of disability [15]. This disability in PwMS is usually assessed through the Kurtzke Expanded Disability Status Scale (EDSS) [16]. On this scale, the ambulatory capacity of the patient has a great weight in the resulting score, but the level of muscular strength is not considered, which is also an important factor for the level of functional independence. Characteristically, MS often affects both sides of the body unevenly, either in sensory or motor functions [17,18].

The aims of this study were to develop reference values of the maximum dynamic (1RM) and isometric (MCVI) forces of the knee extensors for PwMS with different degrees of disability and types of MS, and examine the degree of inter-lower limb asymmetries in force generation.

## 2. Experimental Section

### 2.1. Participants

This observational cross-sectional study was part of a strength training service for PwMS, which was performed by the University of León with the support of the Regional Ministry of Health of the of the Government of Castilla y León (Spain). The research protocol was approved by the Research Ethics Committee at the University of León and data were collected from the first evaluation of muscle strength on the leg extensors carried out in 390 PwMS (149 men and 241 women) of different ages when they joined to the training program. The patient inclusion criteria were confirmed diagnosis of MS based on the McDonald criteria (2001) [19], ability to ambulate with or without help, ability to perform the tests, and that this was the first time they were evaluated in the resistance training program.

Patients made two visits to the laboratory: in the first visit, a medical history was taken, was evaluated the degree of disability based on the EDSS, and a familiarization session was conducted on the weights machine with strength assessment exercises. In the second visit, strength evaluations were performed.

For the analysis of the results the clinical course of MS (relapsing–remitting (RR); primary progressive (PP); secondary progressive (SP)) was taken into account, and the disability status according to the score of the Kurtzke Expanded Disability Status Scale (EDSS), determined by a specialist doctor, (mild = EDSS ≤ 2.5; moderate = EDSS ≤ 5; severe = EDSS ≤ 7.5; and very severe = EDSS > 7.5).

The methods, procedures and data treatments employed in the study were in accordance with ethical standards and the Declaration of Helsinki (revised October 2013). Before the start of the study and after being informed, each patient signed consent to voluntary participation. The research protocol was approved by the Research Ethics Committee at the University of León (30 January 2017, study number 1835). 

### 2.2. Measurement of Outcomes

Maximal voluntary isometric contraction (MVIC) and the one repetition maximum (1RM) test of knee extensors were measured on a multi-station machine (BH^®^ Fitness Nevada Pro-T, Madrid, Spain), both bilaterally (both legs together at once), and unilaterally (each leg singly). Following the protocol used in other studies [20], the length of the lever arm of the machine was adjusted to the length of the leg lower limb, so the patient could push with the front part of the lower leg at the height of the tibial malleolus.

#### 2.2.1. MVIC

The MVIC test was performed with a similar device described in previous reports [20,21,22], and was measured by a load cell (Globus; Codogné, Italy; sample rate 1000 Hz) for 5 s. Data were collected and analyzed with associated software (Globus Ergo Tester v1.5, Codognè, Italy). Participants were seated in an upright position, with a hip joint angle of 110° and a knee joint angle of 90° of flexion. This was measured by the goniometer TEC (Sport-Tec Physio & Fitness, Pirmasens, Germany). During the test, patients were instructed to push as hard as possible from the beginning of the test and then maintain maximal strength against the fixed lever arm of the device for five seconds. Patients were constantly encouraged to perform the exercises during the measurements. Two attempts, separated by one minute around, were carried out by each participant. Only the higher value of both was considered for further analysis.

#### 2.2.2. RM

For maximum dynamic strength evaluation, we follow the protocol of Oliveira et al. (2017) [22]. First, four warm-up repetitions were performed at 50% of the MVIC under the supervision of the trained evaluator and after indicating the patient’s subjective perception of the effort through the OMNI-Resistance Exercise Scale [23]. 

Two repetitions were performed with each load, which progressively increased between 5 and 14 kg, as a function of the reported subjective perception and the quality of technical execution of repeats, with an interval of 3 min between each of the loads. An attempt was made to obtain the 1RM after four or five attempts to avoid excessive fatigue in the subjects, in the few cases where 1RM was not attained in a maximum of 5 attempts, the evaluation was repeated 48 h later. When the patient was able to perform a single repetition, this mobilized load was considered the 1RM.

#### 2.2.3. Asymmetry

Asymmetry scores were calculated for each strength measure (1RM and MCVI) using the following equation:$$\mathrm{Asymmetry}\;\%\;=\left(1-\left(\frac{limb\;strength\;weaker}{limb\;strength\;stonger}\right)\right)\;\times \;100.$$

Therefore, the higher the score, the greater the degree of asymmetry between the extremities [17,24]. 

### 2.3. Statistical Analysis

For statistical analysis, the software SPSS 23.0 for Windows (IBM-Inc, Chicago, IL, USA) was used. For the descriptive statistics, we used the frequency for the categorical variables and quantitative variables were expressed as mean ± SD. Normality was tested by using the Shapiro–Wilk test. Sex comparisons were performed with unpaired t-tests or the non-parametric equivalent, the Mann–Whitney U test, depending on whether the distribution of the variable was normal or not. ANOVA was used to compare the quantitative outcomes among MS type or level of neurological disability with post hoc de Bonferroni, when it was appropriate. Moreover, *p*-value < 0.05 was considered to be statistically significant. The reference values we provide are expressed as cutoff values between quartiles (Q1_Q2; Q2_Q3; Q3_Q4).

## 3. Results

Firstly, Table 1 shows the mean and SD of the main characteristics of the sample; age, body mass index (BMI), years of evolution, and grade of EDSS, differentiated by gender.

Table 2 shows the sample by gender, type of MS, and degree of neurological disability.

Table 3 shows the values of the force outcomes measured by gender.

Table 4 shows values of the assessed variables of the patients in the sample, grouped by type of multiple sclerosis.

In Appendix A are shown Force variables by type of multiple sclerosis by gender.

Table 5 shows the data of the variables studied in the patients grouped by the level of neurological disability. This table does not show the values of the 8 patients (2 males, 6 females) who presented a very severe disability (EDSS > 7.5), since the small number of these patients made it difficult to compare with the other more numerous disability groups.

In Appendix A are shown Force variables by neurological disability level and gender. 

The quartile cutoff scores for the parameters analyzed and grouped according to the evolutionary type of the disease are shown in Table 6.

In Appendix A are shown Quartiles of force variables of sample by type of multiple sclerosis and gender. 

Table 7 shows quartile cutoff scores for measured and grouped variables according to the degree of neurological disability.

In Appendix A are shown Quartiles of force variables of sample by EDSS degree and gender.

Table 8, Table 9 and Table 10 show 1RM and MVIC asymmetry in patients with a value of asymmetry equal or higher to 10%. If asymmetry value was less than 10%, it was considered normal. 

Moreover, Table 8 shows the percentage of patients who present asymmetry above 10% and the average, maximum and minimum values of asymmetry in these patients by gender.

Table 9 shows the percentage of patients who present asymmetry above 10% and the average, maximum and minimum values of asymmetry in these patients by type of multiple sclerosis.

Table 10 shows the percentage of patients who present asymmetry above 10% and the average, maximum and minimum values of asymmetry in these patients by neurological disability level.

## 4. Discussion

This study assessed 390 people with confirmed MS, of which 61.8% were women, as expected since the incidence of this disease is lower in men [25]. MS typically debuts in young adults [26], which explains that although the sample was young (44.5 years in men and 46.3 years in women), the years of disease progression were 9.7 ± 8.2 in men and 9.9 ± 7.5 in women.

Most patients in the sample had RR (60.4% of men and 71% of women); the second most frequent was SP (22.1% of men and 15.7% of women); and the least frequent was PP (17.5% of men versus 13.3% of women). The change in disease course of MS patients is well known, and it is estimated that 80% of patients debut usually with an acute episode with sensory or motor manifestations that in a short period of time disappear without leaving important functional sequelae [27]. However, the repetition of these episodes tends to produce an incomplete recovery, leaving the patient with a certain level of sequelae, is the RRMS form. Over time, about 65% of patients, without presenting flare-ups, evolve more or less rapidly towards a motor or sensory worsening, this evolutionary phase, is the SPMS form. Only a minority of patients (20%) present from the beginning of the disease, without outbreaks, a progressive evolutionary type from the beginning of the disease, is the PPMS form [27]. The pharmacological treatment of the disease is mainly determined by the evolutionary type of the disease; however physical conditioning and rehabilitative treatment are not only determined by the type of disease but by the degree of neurological disability [28,29]. There are different scales to evaluate disability in patients with MS, and the EDSS is the most commonly used in the clinic and in studies with intervention in patients with MS [30], despite its limitations, which include that it gives a very important weight in the final score to the ability to walk, and therefore it prioritizes the evaluation of the lower limbs over the upper ones. The degree of neurological disability (EDSS) does not seem to depend on the sex of the patient, and in our sample men and women have similar EDSS scores. The EDSS is a scale with a result between 0 (minimum disability) and 10 (death from MS), with minimum increments of 0.5 points. There is no consistently accepted categorical grouping of the degree of neurological disability on this scale [31]. For this reason, in order to rationalize the analysis and expression of our results, as well as to increase the practical application of these results by professionals, we have categorized them into four levels by disability status according to the EDSS score: mild = EDSS ≤ 2.5; moderate = EDSS ≤ 5; severe = EDSS ≤ 7.5; and very severe = EDSS > 7.5.

In all people, but more obviously in PwMS, having adequate levels of strength in the lower limbs is fundamental for walking, balance, and the autonomy of the patient [32]. This is one of the reasons why rehabilitation and physical conditioning of these patients is currently focused on strength training of the lower extremities [4,32,33,34,35].

In the evaluation of the performance of a particular muscle group, it is fundamental, especially in the measurement of isometric strength, to standardize the joint angle on which this group acts, as well as that of the joints on which the agonist groups in that exercise are operating [36,37,38]. In general it is considered that the angle of knee flexion which produces the greatest isometric torque is between 80 and 90° [36,39]; however, the angle of the patient’s hip in the evaluation position also modifies the result of the force expressed. Therefore, the standardization of angles is necessary to be able to establish valid comparisons of results obtained in evaluations sequenced over time and for comparison with data from other studies. However, research publications measuring dynamic force or isometric force of knee extensors usually do not indicate hip angles.

This lack of information in manuscripts and sample characteristics of each study make it very difficult to compare our data with those of other published research. Isometric strength assessment is easy and quick to carry out, produces low fatigue in PwMS, shows good repeatability [40], and shows a good correlation with dynamic muscle performance [41], so it is not unusual to find studies in which it has been measured for programming or to see the outcome of a physical training program in patients with MS [20,42].

Women with MS have lower isometric strength values than men, both bilaterally and unilaterally, as would be expected, since also in healthy populations, at any age, men show higher isometric strength values [1,43]. Often studies with PwMS have used isometric strength to prescribe training loads and even as a form of training [33,44], and it has been described that improvement in this outcome after a training programmed is linked with improvements in walking and in the quality of life of those affected by the disease [20,33]. Patients with RRMS show a little more isometric strength than other evolutionary forms of disease, but the difference is small and is influenced not only by the disease, but probably also by age as patients in the RRMS group are younger. Patients with SPMS have more years of disease evolution. To the best of our knowledge no studies compare functional ability according to the type of MS. 

The maximum dynamic force, 1RM, both bilaterally and unilaterally manifested, is also higher in males, but in contrast to the isometric force, it shows no difference between the different evolutionary types of MS. 

An interesting aspect is the study of the degree of symmetry in the lower limbs, as often the disease does not affect both sides of the body equally [45], and PwMS most often present asymmetry in force between limb [17,18,46,47,48,49], and some studies have found that the degree of asymmetry is inversely related to walking ability [24,32,47]. The lower-limb explained a 20–30% of the variance functional capacity tests between PwMS, which is the reason why muscle training is recommended to minimize strength asymmetry between limbs [46], although other studies have not corroborated this association [50]. A certain degree of asymmetry in the strength of the knee extensors can be observed in healthy populations, with up to 10% difference between the extremities considered as normal [51,52]. However, the PwSM in the sample present a level of asymmetry higher than the 10% considered normal, as 70.4% of the men and 60% of the women show asymmetry for MVIC, and for the 1RM 39.5% of the men and 31.2% of the women. Most of the studies that have analyzed force asymmetry in PwMS have done so with isokinetic dynamometry, so the data are not very comparable. Asymmetry in force generation has been more studied in sports populations, probably because of the relationship between asymmetry and sports performance and injury risk [53]. In the physically active or athletic population the degree of isokinetic asymmetry of knee extensors was 3.7%, according to a meta-analysis of seven studies with a total sample of 173 participants [54], and in 259 athletes the average asymmetry in MVIC was 9% [55]. Asymmetry increases significantly after an injury or surgery on the locomotive system and afterwards tends to decrease with physical reconditioning [51]. 

We do not know of any studies that have compared affective asymmetry between patients with different forms of MS; in our study there were no differences between the MS groups for either MRI or MVIC. Accordingly with the studies that have found an association between the degree of disability and the degree of asymmetry [24,32,47], we observed that patients with mild neurological disability have a significantly lower 1RM asymmetry score with respect to patients with moderate disability and in the MVIC with respect to those with severe disability. About 30% of PwSM show an abnormal degree of 1RM asymmetry, and MVIC asymmetry in 34% of patients with mild and 60% of moderate and severe disability.

The studies that have been carried out on symmetry in PwMS have been of a transverse type, so we can only analyze the association of asymmetry with other disease parameters, physiological or functional; longitudinal studies are needed to explain the causal factors of asymmetry, including studies with more standardized methodologies and protocols [18]. For potential use by health or physical activity professionals caring for PwMS, for the variables analyzed in this study we showed reference values with points of separation between quartiles, grouped by type of MS and degree of neurological disability.

Limitations of the study include that the lack of sufficiently standardized protocols in the evaluation of muscle performance in PwMS has made it difficult to compare our data with that of other studies. Nor have we been able to compare the data of our patients with healthy people, since we have not found reference values for these variables in healthy populations of different ages and gender. Due to the inclusion criteria used in this study, there have been few patients with a very severe degree of disability (EDSS > 7.5), so we have not been able to establish comparisons or obtain reference values for them. 

## 5. Conclusions

There is no difference in 1RM according to the disease course of MS, but there is according to the degree of neurological disability, with 1RM more preserved in those with a lower degree of disability. MVIC tends to be higher in patients with relapsing–remitting MS, and in patients with lower levels of neurological disability. Asymmetry above the values considered normal in 1RM is present in between 20 and 60% and MVIC asymmetry in between 56 and 79% of patients, depending on the type of MS, and tends to be lower in those with less disability. Reference values are given by quartiles for 1RM, MVIC, and asymmetry.

## Figures and Tables

**Table 1 ijerph-17-08083-t001:** Characteristics of the sample according to gender.

Variables	Male					Female					
	Mean		SD	Max	Min	Mean		SD	Max	Min	*p*
Age (years)	44.5	±	1.6	75	5	46.3	±	10.8	69	20	0.288
BMI/ kg (m^2^)	25	±	3.6	36.8	9.8	24.5	±	4	36.4	16.7	0.188
Years of evolution	9.7	±	8.2	41	0	9.9	±	7.5	34	0	0.338
EDSS	3.7	±	2.1	8	0	3.5	±	2	8.5	0	0.854

BMI = body mass index; EDSS = Expanded Disability Status Scale; SD = standard deviation.

**Table 2 ijerph-17-08083-t002:** Sample level disability by gender and the type of multiple sclerosis.

Variables	Level	Male				Female			
		PP	RR	SP	Total	PP	RR	SP	Total
Level EDSS	Mild	11	41	5	57	8	82	7	97
	Moderate	10	36	8	54	12	68	17	97
	Severe	4	13	19	36	11	17	13	41
	Very Severe	1	0	1	2	1	4	1	6
	Total	26	90	33	149	32	171	38	241

PP = Primary—Progressive; RR = Relapsing—Remitting; SP = Secondary-Progressive; EDSS = Expanded Disability Status Scale.

**Table 3 ijerph-17-08083-t003:** Strength values by gender.

Variables	Male					Female					
	Mean		SD	Max	Min	Mean		SD	Max	Min	*p*
1RM Bil.	86.8	±	26.8 *	160.0	16.0	58.7	±	21.8	120.0	6.0	0.000
1RM Right	46.9	±	16.5 *	90.0	14.0	28.4	±	10.6	59.0	8.0	0.000
1RM Left	49.3	±	17 *	90.0	21.0	29.4	±	11.4	55.0	0.0	0.000
MVIC Bil.	97.9	±	31.3 *	182.6	24.8	62.9	±	19.6	117.0	16.3	0.000
MVIC Right	46.3	±	16.3 *	101.6	10.0	31.0	±	10.3	62.1	5.4	0.000
MVIC Left	47.2	±	17.4 *	108.6	4.5	30.7	±	10.4	56.5	9.0	0.000
Asymmetry 1RM	10.8	±	14.9	57.6	0.0	11.0	±	17.0	74.2	0.0	0.952
Asymmetry MVIC	20.3	±	16.3	78.4	0.2	17.5	±	15.2	78.6	0.0	0.162

The strength is expressed in kilogram_force (Kg_f). MVIC = maximal voluntary isometric contraction; 1RM = one repetition maximum; Bil. = bilateral; asymmetry is indicated in %; SD = standard deviation; Max = maximum value; Min = minimum value; * = significant difference between genders. Asymmetry is indicated in percentage values.

**Table 4 ijerph-17-08083-t004:** Age, years of evolution, and force variables by type of multiple sclerosis.

Variables	PP(P)					RR (R)					SP (S)					
	Mean		SD	Max	Min	Mean		SD	Max	Min	Mean		SD	Max	Min	*p*
Age	51.8	±	10.9 ^R,S^	70	5	42.1	±	9.7 ^S^	68	20	50.6	±	11.4	75	29	0.000
Years of evolution	6.8	±	7.2 ^R,S^	32	0	9.8	±	7.3	41	0	12.1	±	8.8	34	0	0.002
1RM Bil.	65.4	±	27	120	14	71.8	±	27.2	160	6	61.5	±	27.4	130	12	0.080
1RM Right	31.8	±	15.1	64	13	34.6	±	15.4	90	8	36.8	±	17	77	18	0.653
1RM Left	37.8	±	17.6	75	16	35.9	±	16	90	0	32.8	±	18.1	72	10	0.663
MVIC Bil.	73.5	±	27.8	141.2	27.9	80.7	±	30.5 ^S^	182.6	19.5	65.3	±	29.4	144.7	16.3	0.002
MVIC Right	33.7	±	13.2	65.8	13.5	38.9	±	14.8 ^S^	101.6	10	31.9	±	14.9	70.4	5.4	0.007
MVIC Left	36.2	±	15.7	66.4	9.3	38.7	±	15.9 ^S^	108.6	9	31.4	±	15.1	71	4.5	0.020
Asymmetry 1RM	19.1	±	18.0	55.6	0	9.9	±	15.9	74.2	0.0	6.3	±	10.6	29.0	0.0	0.540
Asymmetry MVIC	20.5	±	13.8	52.0	0.3	16.1	±	14.3 ^S^	78.4	0.0	26.6	±	17.8	74.3	0.5	0.000

The strength is expressed in kilogram_force (Kg_f). MVIC = maximal voluntary isometric contraction; 1RM = one repetition maximum; Bil. = bilateral; asymmetry is indicated in %; SD = standard deviation; Max = maximum value; Min = minimum value. PP = Primary-Progressive; RR = Relapsing–Remitting; SP = Secondary-Progressive. Letters (P, R, S) indicate significant differences between groups. Asymmetry is indicated in percentage values.

**Table 5 ijerph-17-08083-t005:** Age, years of evolution, and force variables by neurological disability level.

Variables	Mild (L)					Moderate (M)					Severe (S)					
	Mean		SD	Max	Min	Mean		SD	Max	Min	Mean		SD	Max	Min	*p*
Age	40.5	±	10.9 ^M,S^	67	5	47.1	±	9.9	75	25	50	±	11.5	64	33	0.000
Years of evolution	8.1	±	7.2S	34	0	10.5	±	7.3	28	0	11.4	±	8.9	41	0	0.019
1RM Bil.	77.5	±	24 ^M,S^	130	23	65.8	±	26.6	160	15	58.6	±	30.8	150	6	0.001
1RM Right	37.9	±	14.8	77	18	32.5	±	16.6	90	8	29.6	±	13.9	55	13	0.100
1RM Left	39.1	±	16.2	85	0	33.3	±	18	90	10	29.2	±	11.2	55	17	0.081
MVIC Bil.	87.7	±	28.1 ^M,S^	179	40	72	±	29.5	176.1	16.3	62.1	±	27.9	161.3	19.2	0.000
MVIC Right	41.9	±	13.6 ^M,S^	102	22.4	35	±	13.6	87	11.1	28.9	±	14.1	65.5	5.4	0.000
MVIC Left	41.3	±	14 ^M,S^	81.6	12	35.5	±	16	89.1	10	29.2	±	13.6	64.7	4.5	0.000
Asymmetry 1RM	8.5	±	10.7	37.3	0.0	11.0	±	19.0	74.2	0.0	13.8	±	18.7	57.6	0.0	0.235
Asymmetry MVIC	13.8	±	11.8 ^M,S^	62.5	0.0	19.5	±	14.7 ^S^	67.7	0.0	26.2	±	18.5	74.3	0.2	0.000

The strength is expressed in kilogram_force (Kg_f). MVIC = maximal voluntary isometric contraction; 1RM = one repetition maximum; Bil. = bilateral; asymmetry is indicated in %; SD = standard deviation; Max = maximum value; Min = minimum value. Letters (L, M, S) indicate significant differences between groups. Asymmetry is indicated in percentage values.

**Table 6 ijerph-17-08083-t006:** Quartiles of force variables of sample by type of Multiple Sclerosis.

Variables	PP			RR			SP		
	Q1_Q2	Q2_Q3	Q3_Q4	Q1_Q2	Q2_Q3	Q3_Q4	Q1_Q2	Q2_Q3	Q3_Q4
1RM Bil.	45.0	61.0	83.0	51.2	68.0	90.0	45.0	55.0	75.0
1RM Right	16.0	30.0	42.0	22.5	33.0	44.5	25.0	27.0	50.0
1RM Left	25.0	35.0	52.0	59.9	73.5	97.3	19.0	30.0	9.0
MVIC Bil.	53.0	66.0	96.5	28.6	37.1	46.0	45.2	56.2	81.0
MVIC Right	22.8	31.7	42.6	28.1	36.0	46.0	22.3	30.3	37.3
MVIC Left	24.7	30.1	51.2	28.1	36.0	47.8	20.1	28.9	39.3
Asymmetry 1RM	20.9	23.5	37.3	15.4	24	34	21.8	26	28.4
Asymmetry MVIC	21.5	25.1	32.4	15.1	20.5	30.6	17.2	28.4	41.1

The strength is expressed in kilogram_force (Kg_f). MVIC = maximal voluntary isometric contraction; 1RM = one repetition maximum; Bil. = bilateral; asymmetry is indicated in %; SD = standard deviation; Max = maximum value; Min = minimum value. PP = Primary-Progressive; RR = Relapsing–Remitting; SP = Secondary-Progressive. Asymmetry is indicated in percentage values.

**Table 7 ijerph-17-08083-t007:** Quartiles of force variables of sample by EDSS degree.

Variables	Mild			Moderate			Severe		
	Q1_Q2	Q2_Q3	Q3_Q4	Q1_Q2	Q2_Q3	Q3_Q4	Q1_Q2	Q2_Q3	Q3_Q4
1RM Bil.	59.0	75.0	90.0	47.0	65.0	75.0	37.5	55.0	78.0
1RM Right	25.0	33.0	46.5	20.0	28.5	42.0	17.2	26.5	42.0
1RM Left	28.0	36.5	49.7	20.0	30.0	42.0	21.0	25.0	35.0
MVIC Bil.	67.1	81.9	100.3	52.8	64.7	78.5	46.0	56.1	80.1
MVIC Right	31.9	39.8	48.5	24.7	31.8	45.1	18.6	27.3	37.5
MVIC Left	31.0	38.7	50.6	23.0	31.5	41.1	19.5	28.9	38.8
Asymmetry 1RM	15.3	20.9	24.7	19	31.5	51	23.5	31.9	36
Asymmetry MVIC	16.1	22.7	27.8	16.6	24.2	35.1	20.1	31.1	44.4

The strength is expressed in kilogram_force (Kg_f). MVIC = maximal voluntary isometric contraction; 1RM = one repetition maximum; Bil. = bilateral; asymmetry is indicated in %; SD = standard deviation; Max = maximum value; Min = minimum value. PP = Primary-Progressive; RR = Relapsing–Remitting; SP = Secondary-Progressive. Asymmetry is indicated in percentage values.

**Table 8 ijerph-17-08083-t008:** Asymmetry of men and women.

Variables	Male					Female					
	Mean	SD	Max	Min	%	Mean	SD	Max	Min	%	*p*
Asymmetry 1RM	25.7	13.5	57.6	10.2	39.5	31.3	17.4	74.2	10.7	31.2	0.295
Asymmetry MVIC	27.1	14.8	78.4	10.6	70.4	26.5	13.9	78.6	10.1	60	0.776

MVIC = maximal voluntary isometric contraction; 1RM = one repetition maximum; asymmetry is indicated in percentage values; % = percentage of patients presenting non-normal asymmetry.

**Table 9 ijerph-17-08083-t009:** Asymmetry of sample by type of multiple sclerosis.

Variables	Multiple Sclerosis Type																		
	PP (P)						RR (R)						SP (S)						
	Mean		SD	Max	Min	%	Mean		SD	Max	Min	%	Mean		SD	Max	Min	%	*p*
Asymmetry 1RM	30.5	±	14.1	55.6	14.7	60	28.2	±	17	74.2	10.2	30	25.1	±	4.3	29	19.4	20	0.845
Asymmetry MVIC	27.6	±	9.5	52.0	10.8	66	24.8	±	13	78.4	10.1	56	30.8	±	16	74.3	10.4	79	0.071

MVIC = maximal voluntary isometric contraction; 1RM = one repetition maximum; asymmetry is indicated in percentage values; % = percentage of patients presenting non-normal asymmetry. PP = Primary-Progressive; RR = Relapsing–Remitting; SP = Secondary-Progressive.

**Table 10 ijerph-17-08083-t010:** Asymmetry of sample by neurological disability level.

Variables	Neurological Disability Level																		
	Mild (L)						Moderate (M)						Severe (S)						
	Mean		SD	Max	Min	%	Mean		SD	Max	Min	%	Mean		SD	Max	Min	%	*p*
Asymmetry 1RM	21.3	±	8.2^M^	37.3	10.2	31	35.9	±	20	74.2	14.7	26	34	±	13	57.6	22.9	35	0.018
Asymmetry MVIC	23.1	±	9.9^S^	62.5	10.2	34	26.4	±	13	67.7	10.1	60	32.9	±	16	74.3	10.7	63	0.004

MVIC = maximal voluntary isometric contraction; 1RM = one repetition maximum; asymmetry is indicated in percentage values; % = percentage of patients presenting non-normal asymmetry.

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
