# Peer review of "Reference Values for Isometric, Dynamic, and Asymmetry Leg Extension Strength in Patients with Multiple Sclerosis"

_ijerph, 2020, doi:10.3390/ijerph17218083_

Round 1

Reviewer 1 Report

Comments for Reference Values for isometric, dynamic and asymmetry leg extension strength in patients with Multiple Sclerosis

General comments: the study examined the one repetition-maximum (1RM) and maximal voluntary isometric contraction (MVIC) values of knee extension exercise in 390 patients with multiple sclerosis (PwMS). Considering the lack of normative reference values in lower limb strength, the topic and the results of the study can be clinically important for evaluating strength training program in PwMS. The authors really need to go through the paper, and proof read the paper very carefully, in addition to the following comments I have below:

Abstract

Line 28, within or in this population

Line 31, “compare the symmetry in the force generation of legs”, change it to “examine knee extension strength asymmetry”

Lines 32-34, “There was no difference in 1RM according to the disease course of MS, but there was according to the degree of neurological disability, being more preserved in those with a lower degree of disability.” I don’t understand what the authors are trying to say in this sentence.

Line 35, “higher in patients with relapsing–remitting MS”, as compared to what?

Lines 36-38, “Asymmetry above the values considered normal in 1RM was present between 20 and 60% and MVIC asymmetry between 56 and 79% of patients depending on the type of MS and tended to be lower in those with less disability.” Again, I don’t follow this sentence here.

Introdcution

Line 62, “Force is the result of active tension muscle, which is…” This sentence doesn’t read. Please rephrase.

Line 66, I would use “different types” or “categories” of muscle strength…

Line 67, the most common type of muscle action.

Line 71, for “1RM”, please provide the full words at the first time it appears: one-repetition maximum (1RM).

Lines 69-71, this sentence is too long, consider break it down.

Line 72 needs reference(s).

Lines 79-81, please rewrite this sentence.

Lines 86, I would reword the sentence to “both sides of the body unevenly, for both sensory and motor functions”

Lines 87-89, this part is too abrupt. The authors haven’t even talked about the rationale of the study: e.g., is there any clinical benefits to establish the reference values for the PwMS? This information is important for readers. In addition, what were authors hypotheses?

Methods

Were participants provided a familiarization prior to any strength testing?

Lines 101-106, are all these belong to the same paragraph? Or different paragraphs?

Line 102-103, the categories of MS need more information (RR vs. PP vs. SP). Can authors provide more information regarding these categories?

For strength testing, more specific details are need. For example, limb testing order? In the abstract (line 32), the authors mentioned “bilateral”, do was any bilateral strength test done? If so, please add the details. The authors also cited references 21-23 for isometric and 1RM strength tests, but a brief description about the testing setups should still be included.

For statistical analysis, was the data normally distributed? Please specify. It will affect which statistical tests you use.

Results and Discussion

The Discussion is a difficult read. There are broken paragraphs all over the places.

I would also like to see a discussion on the values for MS patients vs. normal healthy values.

Author Response

We appreciate your comments regarding the value of the study in the context of lack of data, in the clinical and rehabilitative context of MS patients, and we are grateful for your corrections and consider your observations. Below, we gather point by point indications and we hope we have responded to the observations you have been kind to make.

Thank you very much

Reviewer 1

General comments: the study examined the one repetition-maximum (1RM) and maximal voluntary isometric contraction (MVIC) values of knee extension exercise in 390 patients with multiple sclerosis (PwMS). Considering the lack of normative reference values in lower limb strength, the topic and the results of the study can be clinically important for evaluating strength training program in PwMS. The authors really need to go through the paper, and proof read the paper very carefully, in addition to the following comments I have below:

Point 1: Line 28, within or in this population

Response 1: We have corrected, by replacing in text "among this population" by "in this population” (line 28)

Point 2: Line 31, “compare the symmetry in the force generation of legs”, change it to “examine knee extension strength asymmetry”

Response 2: We have corrected, by replacing in text by "examine knee extension strength asymmetry” (line 30)

Point 3: Lines 32-34, “There was no difference in 1RM according to the disease course of MS, but there was according to the degree of neurological disability, being more preserved in those with a lower degree of disability.” I don’t understand what the authors are trying to say in this sentence.

Response 3: When we say it, we want to say that after analyzing the data there were no statistically significant differences in the 1RM between patients with PP = Primary-Progressive; RR = Relapsing–Remitting o SP = Secondary-Progressive (disease course or type MS,-it's the same thing-); But if there were differences in the 1RM between the disability grade groups ( Mild, Moderate and  Severe  EDSS).

Point 4: Line 35, “higher in patients with relapsing–remitting MS”, as compared to what?

Response 4: We compared patients with RRMS with primary progressive or secondary progressive MS patients. Thus, we added to the paper "respect those with progressive MS". (line 35)

Point 5: Lines 36-38, “Asymmetry above the values considered normal in 1RM was present between 20 and 60% and MVIC asymmetry between 56 and 79% of patients depending on the type of MS and tended to be lower in those with less disability.” Again, I don’t follow this sentence here.

Response 5: The sentence as it is written is indeed unclear. We change this paragraph for this other: “Asymmetry above the values considered normal in 1RM was present in 20-60% of patients and 56-79% in the MVIC test, depending on the type of MS and tended to be lower in those with less disability“ (line 36-38)

Point 6: Line 62, “Force is the result of active tension muscle, which is…” This sentence doesn’t read. Please rephrase.

Response 6: The sentence as it is rewritten more clearly: “There are different types of muscular actions: concentric (muscle length is shortened), eccentric (muscle length is extended), isometric (muscle length is fixed), isotonic (muscle tension is fixed), isokinetic (muscle shortening or lengthening velocity is fixed), and isoinertial (muscle contraction encounters a fixed external resistance), and in all of them muscular tension is generated which we call force”(Line 61-65)

Point 7: Line 66, I would use “different types” or “categories” of muscle strength…

Response 7: we have removed “different expressions of muscle strength” to reformulate the whole paragraph, thus: “There are different types of muscular actions: concentric (muscle length is shortened), eccentric (muscle length is extended), isometric (muscle length is fixed), isotonic (muscle tension is fixed), isokinetic (muscle shortening or lengthening velocity is fixed), and isoinertial (muscle contraction encounters a fixed external resistance), and in all of them muscular tension is generated which we call force” (Line 61-65)

Point 8: Line 67, the most common type of muscle action.

Response 8: We have corrected, by replacing in text "… common type of muscle tension” by " common type of muscle action” (line 66)

Point 9: Line 71, for “1RM”, please provide the full words at the first time it appears: one-repetition maximum (1RM).

Response 9:  We had only placed all the words of the 1RM acronym the first time it appeared in the abstract; we have also added it to the first time it appears in the body of the text. (line 70)

Point 10: Lines 69-71, this sentence is too long, consider break it down.

Response 10:  The sentence has been reformulated and shortened to write as follows: “Traditionally, resistance training is targeted to improve maximum dynamic strength, and the main consensus established on strength training for health, recommends programming the training load based on each person's 1RM”. (line 69-71)

Point 11: Line 72 needs reference(s).

Response 11: we have put as a reference: Med Sci Sport. Exerc 2009, 41, 687–708, and added: J. Neurol. Sci. 2017, 376, 225–241

Point 12: Lines 79-81, please rewrite this sentence.

Response 12: We have reformulated the phrase, writing it as follows: “…But, health and exercise professionals find difficulties in his work for the lack reference data on muscle strength for PwMS with different evolutionary types of the disease or for different degrees of disability.”

Point 13: Lines 86, I would reword the sentence to “both sides of the body unevenly, for both sensory and motor functions”

Response 13: we have substituted the sentence we had written for the one indicated by the reviewer, which is better expressed: “...MS often affects both sides of the body unevenly, either in sensory or motor functions”.

Point 14: Lines 87-89, this part is too abrupt. The authors haven’t even talked about the rationale of the study: e.g., is there any clinical benefits to establish the reference values for the PwMS? This information is important for readers. In addition, what were authors hypotheses?

Response 14:

We have tried to improve this aspect, in writing before the objective of the study, the usefulness of knowing 1RM to prescribe the strength and to have data for comparison with other patients of the same age, degree of disability, and type of disease evolution.

Due to the lack of reference data in the scientific and clinical community for the dynamic strength of knee extensors, (an important muscle group for daily life activities), in patients with MS, the aim of this work was to contribute to provide data from a wide sample of these patients, so there was no starting hypothesis.

With the main objective of the study, to offer reference data, the previous explanatory formulation of a hypothesis may not be particularly important, which is why we did not formulate it.

Point 15: Were participants provided a familiarization prior to any strength testing?

Response 15: We had not included it in the initial document; we have added the most detailed explanation, writing the paragraph as follows; Patients made two visits to the laboratory: in the first visit, a medical history was taken, was evaluated the degree of disability based on the EDSS, and a familiarization session was conducted on the weights machine with strength assessment exercises. In the second visit, strength evaluations were performed”

Point 16: Lines 101-106, are all these belong to the same paragraph? Or different paragraphs?

Response 16: were in fact written in a very disconnected way. We have rewritten the paragraph as follows: For the analysis of the results the clinical course of MS (relapsing–remitting, RR; primary progressive, PP; secondary progressive, SP) was taken into account, and the disability status according to the score of the Kurtzke Expanded Disability Status Scale (EDSS), determined by a specialist doctor, (mild = EDSS ≤ 2.5; moderate = EDSS ≤ 5; severe = EDSS ≤ 7.5; and very severe = EDSS > 7.5)”

Point 17: Line 102-103, the categories of MS need more information (RR vs. PP vs. SP). Can authors provide more information regarding these categories?

Response 17: We had placed that explanation at the beginning of the discussion, to make the reading of the discussion more friendly, for readers not familiar with this disease. We have touched up the wording by leaving the explanation like this (with a bibliographic reference): The change in disease course of MS patients is well known, and it is estimated that 80% of patients debut usually with an acute episode with sensory or motor manifestations that in a short period of time disappear without leaving important functional sequelae. However, the repetition of these episodes tends to produce an incomplete recovery, leaving the patient with a certain level of sequelae, is the RRMS form. Over time, about 65% of patients, without presenting flare-ups, evolve more or less rapidly towards a motor or sensory worsening, this evolutionary phase, is the SPMS form. Only a minority of patients (20%) present from the beginning of the disease, without outbreaks, a progressive evolutionary type from the beginning of the disease, is the PPMS form”

Point 18: For strength testing, more specific details are need. For example, limb testing order? In the abstract (line 32), the authors mentioned “bilateral”, do was any bilateral strength test done? If so, please add the details.

Response 18: we have added for clarity: tests of bilateral (both legs together at once) and unilateral (each leg singly) knee extensors” (line 31-32). We have also added the order of the tests: “The evaluation was performed in exactly this order: MVIC: bilateral, leg right, leg left; 1RM: bilateral, leg right, leg left.” (line 116-117).

Point 19: The authors also cited references 21-23 for isometric and 1RM strength tests, but a brief description about the testing setups should still be included.

Response 19: In addition to the references, we had already indicated the detailed procedure of the evaluation of the MVIC (line 119-129) and the 1RM (131-140). We have added in the 1RM procedure the range of increase between each load and the rest time between each new load.

Point 20: For statistical analysis, was the data normally distributed? Please specify. It will affect which statistical tests you use.

Response 20: In fact, we did study the normality of the distribution, and we had indicated this in section 2.3 (lin151). We also indicated in the description of the statistical analysis that for the comparison of means T-Test or Mann-Whitney U was used. With this sentence, it is implicit that depending on whether it was normal or not, parametric or non-parametric measures were used. For more clarity we have added: “ …depending on whether the distribution of the variable was normal or not.” (lin 153).

And ANOVA, is a robust test to the lack of normality, (http://www.psicothema.com/pdf/4434.pdf;  https://statistics.laerd.com/statistical-guides/one-way-anova-statistical-guide-3.php#:~:text=The%20one%2Dway%20ANOVA%20is,its%20normality%20assumption%20rather%20well.)

The expression of the results in quartiles (or percentiles) is not affected by the normality or not of the distribution of the variables, since it only expresses the order of the components of the sample according to the magnitude of the values.

The values of force, did not present a normal distribution, in general the average and the mode were very similar values, but not the median because there was an elongated distribution by one of the tails (the one with higher values). In other words, it did not completely fulfil the criteria of normality, but it was near to a normal distribution, only elongated by one of the tails of the distribution.

Point 21: The Discussion is a difficult read. There are broken paragraphs all over the places.

Response 21: We have eliminated some isolated paragraphs, and we have tried to reorder the discussion and introduced some paragraphs, to make this section even easier to read decrease to the degree that we have been able to broken paragraphs. The discussion has a sequence that is: type of MS, degree of EDSS, why evaluate strength and the importance of standardization, isometric strength, 1RM, and symmetry.

Point 22: I would also like to see a discussion on the values for MS patients vs. normal healthy values.

Response 22: It was outside the scope of the work the comparison of functional capacity with respect to the healthy population of the same age and gender; they are clearly different populations with evident and different functional capacity, this was the reason why in the design it was not considered to search healthy population of similar age and gender characteristics. We do not know yet publications of reference values of 1RM and MVCI in healthy population (yes of with isokinetic evaluations, but not with isoinerciales), with similar methodology to that of our study. There are articles with particular populations in which 1RM and/or MVCI have been measured in the context of the study, with a similar methodology to the one we have used, with the values presented in those specific studies we could compare, but it would be to compare with an unrepresentative sample of the general healthy population. Not being capable of adding to the discussion with sufficient precision this aspect pointed out by the reviewer, we have added it as one of the limitations of the study: “Nor have we been able to compare the data of our patients with healthy people, since we have not found reference values for these variables in healthy populations of different ages and gender.   “ (line 349-350)

Reviewer 2 Report

In document you explain that an attempt was made to obtain the 1RM after four or five attempts to avoid excessive fatigue in the subjects. What happened in those cases where it was not possible to obtain the 1RM without fatigue? Was the failure test performed? I believe that this should be well detailed in the text.
 In the case of a population not necessarily trained, who do not know their approximate 1RM, it might have been convenient to obtain this data by measuring the speed of movement of the exercise to estimate its maximum repetition. This has been shown to be a more precise way and superior to other methods of estimating 1RM, since the speed of movement is highly correlated with the strength capabilities of the subject.

You have registered improvements in walking and quality of life. Given that the asymmetry above the values ​​considered normal in 1RM is present between 20 and 60% and the asymmetry of the MVIC between 56 and 79% of patients according to the type of MS and tends to be lower in those with less disability, perhaps it would have been interesting if you also registered the risk of falls.

It is very interesting that you have compared functional capacity according to the type of MS. I consider the study very interesting, although others with standardized methodologies and protocols are necessary. With these results, strength and resistance treatment protocols could be carried out by health and physical activity professionals at PwMS.

Author Response

Response to Reviewer 2 Comments

We appreciate your comments and we are grateful for your corrections and consider your observations. Below, we gather point by point indications and we hope we have responded to the observations you have been kind to make.

Thank you very much

Reviewer 2

Comments and Suggestions for Authors

Point 1: In document you explain that an attempt was made to obtain the 1RM after four or five attempts to avoid excessive fatigue in the subjects. What happened in those cases where it was not possible to obtain the 1RM without fatigue? Was the failure test performed? I believe that this should be well detailed in the text.

Response 1: We have clarified this point, added: “in the few cases where 1RM was not attained in a maximum of 5 attempts, the evaluation was repeated 48 hours later”.

Point 2: In the case of a population not necessarily trained, who do not know their approximate 1RM, it might have been convenient to obtain this data by measuring the speed of movement of the exercise to estimate its maximum repetition. This has been shown to be a more precise way and superior to other methods of estimating 1RM, since the speed of movement is highly correlated with the strength capabilities of the subject.

Response 2: This aspect of predicting 1RM based on the speed of load movement is a relatively recent development. It is based on changes in the load/load displacement speed ratio, and the speed at which the 1RM load is executed. But these data are not yet sufficiently studied in patients with MS, so if we had used it that prediction would have to be based on data from other healthy populations, young or old, or in any case populations without MS. In order to use this method for 1RM estimation, it is necessary to use a position transducer (encoder) or applications with video capability to determine the displacement velocity. It was out of our goal, and out of the initial possibilities we had. It will be very interesting to analyze this suggestion proposed by the reviewer in future studies carried out with PwMS.

In any case, we have not estimated 1RM, but have determined it directly. We used the MCVI value for the determination of the heating load, as we have previously published, given the correlation between isometric force and dynamics.

Point 3: You have registered improvements in walking and quality of life. Given that the asymmetry above the values ​​considered normal in 1RM is present between 20 and 60% and the asymmetry of the MVIC between 56 and 79% of patients according to the type of MS and tends to be lower in those with less disability, perhaps it would have been interesting if you also registered the risk of falls.

Response 2: Certainly it would have been interesting to ask patients about this specific and important aspect, but unfortunately we did not do so at the time. We will do it from now on. We do have studies with stabilometry, but not in the sample of this work, https://pubmed.ncbi.nlm.nih.gov/28878471/

Point 3: consider the study very interesting, although others with standardized methodologies and protocols are necessary. With these results, strength and resistance treatment protocols could be carried out by health and physical activity professionals at PwMS.

Response 3: The methodology of the direct determination of 1RM, is a fundamentally standardized methodology, there are more than 32 good quality published studies on the repeatability of this peak force determination (https://pubmed.ncbi.nlm.nih.gov/32681399/ ), and used in many populations including PwMS, where it began to be used around 2005  (https://pubmed.ncbi.nlm.nih.gov/16966232/.), at least in a healthy population, presents a high repeatability, but the reviewer is right that there is also a gap in the knowledge of 1RM repeatability in PwMS, that will certainly be interesting to explore. There is less and less fear, outside the hospital environment, to evaluate directly the force in any type of population and it is used more and more frequently.

Reviewer 3 Report

Thank you for the opportunity to review this manuscript, which considers some interesting, applied issues. This study appears to be novel, but as submitted needs considerable work on the presentation. The authors showed an interesting point about the “Reference Values for isometric, dynamic and asymmetry leg extension strength in patients with Multiple Sclerosis”, unfortunately, there are several points to overcome.

Major points

This investigation is very interesting but the main document showed several flaws:

The: “Maximal voluntary isometric contraction (MVIC) and the one repetition maximum (1RM) test of knee extensors was measured on a multi-station machine (BH® Fitness Nevada Pro-T, Madrid, Spain). Following the protocol used in other studies” but this method seem to be not validated, furthermore several details are missing (Sample rate, precision and accuracy)

The authors don’t calculated the bilateral Deficit (this is a big news) https://pubmed.ncbi.nlm.nih.gov/24346189/

A Strong statistical analysis (thus: ANOVA with repeated measured in-between groups) and test-retest are missing

Minor point

The U.M. are missing in the tables

Author Response

Response to Reviewer 3 Comments

We appreciate your comments and we are grateful for your corrections and consider your observations. Below, we gather point by point indications and we hope we have responded to the observations you have been kind to make.

Thank you very much

Reviewer 3

Comments and Suggestions for Authors

Thank you for the opportunity to review this manuscript, which considers some interesting, applied issues. This study appears to be novel, but as submitted needs considerable work on the presentation. The authors showed an interesting point about the “Reference Values for isometric, dynamic and asymmetry leg extension strength in patients with Multiple Sclerosis”, unfortunately, there are several points to overcome.

Major points

This investigation is very interesting but the main document showed several flaws:

Point 1: The: “Maximal voluntary isometric contraction (MVIC) and the one repetition maximum (1RM) test of knee extensors was measured on a multi-station machine (BH® Fitness Nevada Pro-T, Madrid, Spain). Following the protocol used in other studies” but this method seem to be not validated, furthermore several details are missing (Sample rate, precision and accuracy)

Response 2: We believe that the main aspects of the validation of these methods (MVIC and 1RM), have been carried out and published, there are more than 32 good quality published studies on the repeatability of this peak force determination (https://pubmed.ncbi.nlm.nih.gov/32681399/ ), and used in many populations, and the scientific publications in which the determination of 1RM and MVIC has been used are innumerable (in pubmed 740 publications with "MVIC", and 2,894 with "1RM"; in both cases with articles on validity and repeatability. There are also some meta-analyses such as the one that includes 32 analyzed studies https://pubmed.ncbi.nlm.nih.gov/32681399/). It is true, as the reviewer remarks, that there are far fewer in the PwMS population and as the reviewer suggests it would be appropriate to conduct specific repeatability and validity studies in the MS population. The methods are sufficiently validated, not so much their use in PwMS, but it is quite used. With sufficiently contrasted methods, it is not usual to carry out particular validation studies; for example, VO2max studies on diabetics or cardiac patients are used and published, without the validity of their use in that specific population having been analyzed in those studies or previously, although it is clear that it is appropriate to carry them out.

Point 2: The authors don’t calculated the bilateral Deficit (this is a big news) https://pubmed.ncbi.nlm.nih.gov/24346189/

Response 2: It is undoubtedly an interesting aspect of study at PwMS. But that aspect deserves a complete specific work, and escapes the objectives of this work. We have explored this field with both healthy and some PwMS data, and our data are very inconsistent about whether or not there is a congruent bilateral deficit, and whether the differences, although significant, are larger than the minimally detectable change by the method. Interestingly, it should be considered for future studies.

Point 3: A Strong statistical analysis (thus: ANOVA with repeated measured in-between groups) and test-retest are missing

Response 3: We believe that it is not possible to carry out the analyses proposed by the reviewer in this study. The study is cross-sectional, each individual or group of patients was deprived of strength on a single occasion, so that no repeated measurements, test-retest or factor analysis can be determined. That is why we have used single-factor ANOVA (degree of disability and type of MS). For the analysis of the interaction of the two factors, we do not have enough statistical power, but no doubt it will be interesting to study it.

Point 3: The U.M. are missing in the tables  

Response 3: the units of measurement of the force are collected in each of the table foot. In all cases it has been expressed as kg, which as it is strictly a unit of mass, we have indicated kilogram_force (kg_f).

Round 2

Reviewer 1 Report

Thank you for the revisions of this manuscript. Also, I am satisfied with the responses you had to my previous comments.

Author Response

We are pleased to have made the indications you requested and to answer your comments. We sincerely appreciate all your corrections and observations that have helped to improve the manuscript. 

Reviewer 3 Report

This investigation showed several flaws with big limitations

Author Response

We accept your assessment, although we do not give you any answer, as you have not made any specific objections or changes.

There are many of us who act as referees, (me this year on 16 occasions), and it is an ungrateful and fundamental work for the selection and improvement of what is published. So first of all I want to thank you for your time. But if you allow me, with all the respect you deserve, and from colleague to colleague, I would have liked to read the arguments of your objections, among other reasons because they would have helped me to improve.

A grateful greeting.